# Enzyme Properties of a Laccase Obtained from the Transcriptome of the Marine-Derived Fungus *Stemphylium lucomagnoense*

**DOI:** 10.3390/ijms21218402

**Published:** 2020-11-09

**Authors:** Wissal Ben Ali, Amal Ben Ayed, Annick Turbé-Doan, Emmanuel Bertrand, Yann Mathieu, Craig B. Faulds, Anne Lomascolo, Giuliano Sciara, Eric Record, Tahar Mechichi

**Affiliations:** 1Biodiversité et Biotechnologie Fongiques, Aix-Marseille Université, INRAE, UMR1163 Marseille, France; amal.benayed@enis.tn (A.B.A.); annick.doan@inrae.fr (A.T.-D.); Emmanuel.Bertrand@Univ-Amu.Fr (E.B.); Craig.Faulds@Univ-Amu.Fr (C.B.F.); anne.lomascolo@univ-amu.fr (A.L.); giuliano.sciara@inrae.fr (G.S.); eric.record@inrae.fr (E.R.); 2Laboratoire de Biochimie et de Génie Enzymatique des Lipases, Ecole Nationale d’Ingénieurs de Sfax, Université de Sfax, Sfax 3029, Tunisia; tahar.mechichi@enis.rnu.tn; 3Michael Smith Laboratories, University of British Columbia, Vancouver, BC V6T 1Z4, Canada; yann.mathieu85@gmail.com

**Keywords:** laccase, *Stemphylium*, heterologous expression, enzyme properties, alkaline, salt tolerance

## Abstract

Only a few studies have examined how marine-derived fungi and their enzymes adapt to salinity and plant biomass degradation. This work concerns the production and characterisation of an oxidative enzyme identified from the transcriptome of marine-derived fungus *Stemphylium lucomagnoense*. The laccase-encoding gene *Sl*Lac2 from *S. lucomagnoense* was cloned for heterologous expression in *Aspergillus niger* D15#26 for protein production in the extracellular medium of around 30 mg L^−1^. The extracellular recombinant enzyme *Sl*Lac2 was successfully produced and purified in three steps protocol: ultrafiltration, anion-exchange chromatography, and size exclusion chromatography, with a final recovery yield of 24%. *Sl*Lac2 was characterised by physicochemical properties, kinetic parameters, and ability to oxidise diverse phenolic substrates. We also studied its activity in the presence and absence of sea salt. The molecular mass of *Sl*Lac2 was about 75 kDa, consistent with that of most ascomycete fungal laccases. With syringaldazine as substrate, *Sl*Lac2 showed an optimal activity at pH 6 and retained nearly 100% of its activity when incubated at 50°C for 180 min. *Sl*Lac2 exhibited more than 50% of its activity with 5% wt/vol of sea salt.

## 1. Introduction

The laccases (benzenediol:oxygen oxidoreductase, EC 1.10.3.2) belong to a small group of enzymes called the blue copper protein or copper oxidases [1] that catalyse the one-electron oxidation of four reducing-substrate molecules concomitant with the four-electron reduction of molecular oxygen to water [2]. Laccases are almost ubiquitous enzymes, widely distributed among plants, fungi, prokaryotes, and arthropods [3]. Before 2011, more than 100 laccases from Basidiomycota and Ascomycota were purified and characterised [4]. They are characterised by a molecular mass and an isoelectric point (pI) that range from about 50 to 100 kDa and 3 to 7, respectively [5,6]. Laccases have active sites containing four copper atoms (Cu) bound to three redox sites (Type 1, Type 2, and Type 3 Cu pair) involved in the catalytic mechanisms of these cuproproteins [7,8]. Christopher et al. [9] identified three types of copper using UV/visible and electronic paramagnetic resonance (EPR) spectroscopy. Type 1 Cu at its oxidised resting state is responsible for the blue colour of the protein and is EPR-detectable. Type 2 Cu is only EPR-detectable [9]. Type 3 Cu is composed of a pair of Cu atoms in a binuclear conformation, which is not detectable by EPR. The Type 2 and Type 3 copper sites form a trinuclear centre directly involved in the enzyme catalytic mechanism [9]. Laccases exhibit broad substrate ranges that vary from one laccase to another [10]. They are known as *p*-diphenol:oxygen oxidoreductases, preferentially oxidising monophenols such as 2,6-dimethoxyphenol or guaiacol. Laccase-catalysed reactions are strongly dependent on redox potential, temperature and reaction medium [11]. Laccases fall into two groups depending on their redox potential. Low-redox-potential enzymes occur in bacteria and plants, and high-redox-potential laccases are widely distributed in fungi [12,13]. The low-redox laccases (400–600 mV), unlike high-redox potential laccases (700–800 mV) and ligninolytic peroxidases (>1 V), enable only direct degradation of phenolic compounds of low redox potential and not oxidation of more recalcitrant aromatic compounds, such as some industrial dyes [14]. In the presence of a suitable redox mediator, for example 1-hydroxybenzotriazole (HBT), laccases are able to oxidise non-phenolic structures [15,16]. Many industrial applications for laccases have been proposed, including in pulp and paper [17], organic synthesis, environment, food, pharmaceuticals, and textile dye decolourisation [18].

Synthetic dyes are chemicals widely used in many industries including textiles, paper, printing, cosmetics, and pharmaceuticals [19]. The discharge of their effluents into water systems is a serious environmental concern [20,21]. Enzymatic decolourisation of dyes has shown several advantages over the physico-chemical methods, including low energy costs, ease of control, and eco-friendly impact on ecosystems [22]. However, biological treatments of dye wastewater usually involve an environment with high salinity and numerous organic solvents. Most laccases lose activity under these extreme conditions [23]. It is therefore important to identify new laccases with high tolerance to salt, organic solvents, and high temperature [24].

According to Bonugli-Santos et al. [25], studies of laccases from marine-derived fungi are still limited, and these enzymes may have different properties from those produced by terrestrial microorganisms, owing to different environmental conditions, such as salinity, temperature, pH and pressure. *Stemphylium lucomagnoense* was previously selected by screening fungi isolated from Tunisian coastal waters for its capacity to exert a laccase-like activity under saline conditions [26]. In addition, its secretome analysis confirmed the presence of laccase-like activities in the presence of sea salts [27]. In this study, we expressed a laccase encoding gene from *S. lucomagnoense* in *Aspergillus niger* and purified and characterised the corresponding protein to study the main properties of this marine-derived enzyme.

## 2. Results

### 2.1. Target Selection, Aspergillus Niger Transformation, and Screening

Among the seven putative full-length laccase-encoding genes annotated from the transcriptome of *S. lucomagnoense* [27] (NCBI BioSample accession ID:191 SAMN15897915 with corresponding NCBI BioProject accession ID: PRJNA659110, assembled contig sequences available at GitHub URL as 203 https://github.com/drabhishekkumar/Stemphylium-lucomagnoense-transcriptomics), one was selected for heterologous expression in *A. niger*, based on the divergence (i) of the predicted protein sequence and (ii) of the predicted pI compared with those of known laccases, and on (iii) the presence of the coordination site for Type 1, 2, and 3 coppers, as found in already-characterised laccases (Table 1 and Appendix A). The full-length gene encoding *Sl*Lac2 consisting of 1839 bp (613 amino acids including 21 amino acid for the signal peptide) was selected for the following reasons. *Sl*Lac2 shared only 44.1, 44.8, and 44.6% identity with the closest laccases *Melanocarpus albomyces* (*Ma*Lac), and *Pestalotiopsis* sp. *Ps*Lac1 and *Ps*Lac2, respectively. The calculated molecular weight was 64.6 kDa with a theoretical pI of 8.07, 3.4 to 2 points higher than the lowest and highest calculated pI related to APMZ2_prot2393 (pI 4.66) and APMZ2_7477 (pI 6.03) proteins. In comparison with the closest characterised laccases, *Ps*Lac1, *Ps*Lac 2, and *Ma*Lac have a calculated pI of 6.17, 4.13, and 5.13, respectively (accession numbers KY55480, KY554801, and Q70KY3, respectively). The theoretical extinction coefficients of *Sl*Lac2 at 280 nm was 111,645 M^−1^ cm^−1^.

To produce the recombinant *Sl*Lac2, protoplasts from *A. niger* D15#26 were co-transformed with a mixture of plasmid pAB4-1 and an expression vector containing the corresponding gene. Transformants were selected for their ability to grow without uridine supplementation, and co-transformants containing the laccase-encoding cDNA were screened for laccase expression by growth on minimum medium plates supplemented with 2,2′-azino-bis (3-ethylbenzothiazoline-6-sulphonic acid) (ABTS). Laccase-producing transformants were identified by the appearance of a green zone around the colonies after 3 to 6 days of culture. Coloured zones on plates were not observed for control transformants lacking the laccase-encoding cDNA (transformed only with pAB4-1). A total of 27 positive clones were cultured in standard liquid media and checked daily for protein production by sodium dodecyl sulfate-polyacrylamide gel electrophoresis (SDS-PAGE) and for enzymatic activity by spectrophotometric assays. Approximately 90% of the tested transformants exhibited laccase-like activity in the culture medium and the transformant *Sl*Lac2 reached a peak of laccase activity (85.9 nkat mL^−1^) on Day 9.

### 2.2. Purification of the Recombinant Laccase and Study of Physico-Chemical Properties

The recombinant laccase was purified from the culture medium of *A. niger* in a three-step procedure (Table 2): ultrafiltration, anion-exchange chromatography, and size exclusion chromatography. 930 mL of the filtered culture media (containing 176 mg of protein) was concentrated (1.9-fold) and separated from most impurities, which included a brown pigment absorbing strongly at 280 nm [28], by ultrafiltration through a polyethersulphone membrane (10 kDa molecular mass cut-off), with a resulting purification factor of 1.4-fold. The resulting concentrate was then loaded onto a carboxymethyl (CM)-Sepharose column to be further purified with a purification factor of 3-fold. In the last step, the resulting sample was concentrated through a 10 kDa molecular mass cut-off Amicon membrane and loaded onto a Sephacryl S-200HR size exclusion chromatography column. The resulting purification factor was 5.9-fold, yielding 7.2 mg of protein. The final recovery yield was 24.1%. The molecular mass of the purified laccase by SDS/PAGE was about 75 kDa (Figure 1A).

To check the N-terminal protein processing, the first five amino acids of the recombinant laccase were sequenced and aligned with the deduced mature protein. They showed 100% identity. This result demonstrated that the 24-amino-acid glucoamylase (GLA) prepropeptide from *A. niger* was correctly processed. For N-glycosylation, five sites (71, 115, 232, 404, and 424) were predicted. In addition, *Sl*Lac2 was deglycosylated using N-glycosidase PNGase (Figure 1B). On SDS-PAGE, the deglycosylated protein showed an apparent difference in its molecular mass from the untreated protein. The molecular masses of the purified and digested laccases were 75 kDa and <70 kDa, respectively. The molecular mass of the deglycosylated *Sl*Lac2 and the corresponding calculated molecular mass (64.6 kDa) confirmed the presence of N-glycosylations of approximately 10% of the total protein.

### 2.3. Kinetic Parameters

The kinetic constants (*K*_M_ and *V*_max_) of *Sl*Lac2 were determined using various substrates such as 2,2′-azino-bis (3-ethylbenzothiazoline-6-sulphonic acid) (ABTS), 2,6-dimethoxyphenol (DMP) and syringaldazine under the optimal reaction conditions at 30 °C and pH (4.0, 5.0 and 6.0 respectively) (Table 3). Of the six substrates, *Sl*Lac2 exhibited the highest affinity for syringaldazine (*K*_M_ = 0.0035 mM) followed by ABTS (*K*_M_ = 0.0206 mμM) and DMP (*K*_M_ = 0.024 mμM). The highest catalytic efficiency was found for ABTS (7.42 s^−1^ mM^−1^), followed by syringaldazine (2.11 s^−1^ mM^−1^) and DMP (0.24 s^−1^ mM^−1^).

### 2.4. Enzyme Activity and Stability at Different pH and Temperature Values

*Sl*Lac2 activity increased gradually with increasing temperature up to 60 °C, the optimal temperature value (Figure 2A). Regarding thermal stability, *Sl*Lac2 was stable at temperatures ranging from 25 to 50 °C, but the activity was significantly lost above 60 °C (Figure 2B). The half-life value for *Sl*Lac2 was 95 min and 30 min, at 60 °C and 70 °C, respectively. The highest laccase activity level was recorded at pH 6.0 (Figure 2C). The enzyme possessed significant activity at pH 4.0 and 7.0 (60% and 75%, respectively). A lower activity could be measured at alkaline pH, with 30% of its initial activity at pH 8.0, and the activity almost disappeared above pH 8.0. When incubated at pH 5.0, 6.0 and 7.0 (Figure 2D), laccase activity of the purified enzyme *Sl*Lac2 retained around 80% initial activity after 4 h of incubation, about 70% after 24 h and 60% after 48 h. When incubated at pH 3.0 or 4, the laccase activity retained over 65% residual activity after 4 h, about 50–65% after 24 h and about 50–60% after 48 h. These results demonstrate that the enzyme endures at neutral to weak-acidic pH but has lower resistance to strong-acidic pH values.

### 2.5. Effect of Metal Ions, Inhibitors and Solvents on Laccase Activity

Various metal ions (Mo^2+^, Ag^2+^, AsO_4_^3−^, Fe^2+^, Cd^2+^, Zn^2+^, and Cu^2+^) at 10 mM concentration were tested on the activity of *Sl*Lac2 (Figure 3A). Mo^2+^ and Ag^2+^ caused the disappearance of 30% and 60% of *Sl*Lac2 activity, respectively. AsO_4_^3^, Fe^2+^, Cd^2+^, and Zn^2+^ led to an even greater decrease in laccase activity (about 90% inhibition), and Cu^2+^ completely inhibited *Sl*Lac2 activity (98%).

Three concentrations (0.05 to 0.5 mM) of potential laccase inhibitors, ethylenediamine tetra-acetic acid (EDTA), sodium dodecyl sulfate (SDS), 2-mercaptoethanol, cysteine, and sodium azide (NaN_3_) were evaluated to check their effects on the *Sl*Lac2 activity (Figure 3B). In the tested concentration range, more than 85% of enzyme activity remained in the presence of EDTA and SDS. Sodium azide inhibited *Sl*Lac2 with a loss of 25–70% of its initial activity, while cysteine acted as a stronger inhibitor (15–85% of inhibition). The most efficient inhibitor was found to be 2-mercaptoethanol, which reduced activity by 88–91%.

The effect of different concentrations of solvents ethanol, methanol, isopropanol, glycerol and acetone on *Sl*Lac2 activity in the range 10–40% (vol/vol) was studied (Figure 3C). *Sl*Lac2 was relatively stable, with 10–20% (vol/vol) of ethanol, methanol and isopropanol, but its activity was increased to more than 130% of its initial activity with 20% (vol/vol) ethanol. However, the recombinant laccase was less stable towards 40% (vol/vol) of these solvents, with a decrease of about 20% of its activity (methanol) to 60% (isopropanol). Of all the organic solvents tested, acetone showed a markedly negative effect on *Sl*Lac2 activity, causing more than 80% activity inhibition at a concentration of 10% (vol/vol) and near-inactivation with 20% and 40% (vol/vol) of acetone.

### 2.6. Effect of Sea Salt on Laccase Activity and Surface Charge of SlLac2

The effect of different concentrations of sea salt (1–5% wt/vol) on the purified laccase was analysed (Figure 4). The results showed that the activity gradually decreased as the salt concentration increased from 10% to almost half of its initial activity (Figure 3A). The enzyme could thus tolerate the presence of sea salt, retaining more than 50% of its activity with 5% wt/vol of sea salt. In parallel, analysis of the primary sequence and the overall surface charges of *St*Lac2 were carried out and compared with those of the terrestrial-derived laccase, *Ma*Lac1, and of the marine-derived laccases, *Ps*Lac1 and *Ps*Lac2. The primary sequence of *St*Lac2 exhibited a similar recurrence of negatively charged (D + E) over positively charged (R + K) amino acids, with the exception of *Ps*Lac2, which had 4.15 times higher (D + E)/(R + K) ratio (Figure 4B). *Ma*Lac1 and *Ps*Lac1 presented intermediate values, i.e., 1.55 and 1.20, respectively. The three-dimensional models of *Ma*Lac1 and *Ps*Lac1 showed an even charge distribution in their surface plots, whereas for *Ps*Lac2, the surface charge was strongly negative (Figure 4C). In contrast, *St*Lac2 exhibited a slightly higher positive charge distribution at the surface.

## 3. Discussion

The objective of our research project was to seek new insights into the physiology of the marine-derived fungi, more specifically their lignocellulose enzyme machinery, and how they adapt in their marine environment. In previous work, we screened five fungal strains isolated from Tunisian coastal waters [26] and characterised for their laccase-like activities. Based on a microbial approach and on enzyme activity screening, one of these marine-derived strains, *S. lucomagnoense*, was selected for its adapted growth on xylan in saline conditions, and its improved laccase (seagrass-containing cultures) and cellulase (wheat straw-containing cultures) activities in the presence of sea salt [27]. To further study the selected marine-derived fungus, its transcriptome was sequenced and its proteome analysed when grown on wheat-straw and sea grass in the absence or presence of sea salt. From this study, we were unable to identify the laccases involved in the improved laccase activity in the presence of salts, possibly owing to the sensitivity limit of our combined transcriptomic-proteomic approach, because of undetected transcript domains. However, seven genes were identified in *S. lucomagnoense* transcriptome that encoded putative laccases, and one was selected for heterologous expression in *A. niger* and characterisation of the corresponding laccase, *Sl*Lac2.

Only a few biochemical characterisations of laccases from marine organisms are available [28,29,30,31]. Our aim was accordingly to obtain additional data on these enzymes and their potential for biotechnological applications. Marine-derived fungi are considered as a source of original enzymes active in saline conditions and pH ranging from 3.0 to 11.0 [24], which are properties of interest for many industrial applications. *Sl*Lac2 was produced in *A. niger* with a yield of about 30 mg L^−1^ in the culture medium, which is in the range of protein production obtained for the laccase of the terrestrial fungus *Pycnoporus cinnabarinus*, heterogously produced in *A. niger* (70 mg L^−1^) [32]. Recently, two laccase-encoding genes isolated from the marine-derived fungus *Pestalotiopsis* sp., were expressed in *A. niger* and lower yields were obtained: 2.6 and 6.2 mg L^−1^ for *Ps*Lac 1 and *Ps*Lac2, respectively [28]. The main physico-chemical properties of *Sl*Lac2 were determined to compare them with literature reports on laccases from marine and terrestrial fungi. The purified laccase was active on all typical laccase substrates including ABTS, DMP, and syringaldazine, and showed the highest affinity (lowest *K*_M_ value) for syringaldazine (3.5 μM). However, the highest catalytic efficiency (*K*_cat_/*K*_M_) was found for ABTS (7.42 s^−1^ mM^−1^) as its turnover value was much higher (*K*_cat_ 0.153 s^−1^) compared with that determined for syringaldazine (0.0074 s^−1^). Laccases are widespread in the natural environment, and very well-characterised for their kinetic parameters. Many previous studies found that syringaldazine and ABTS were their preferred substrates [33]. Laccase affinity for syringaldazine (tens of µM) is generally higher than that measured for ABTS (hundreds of µM), whereas *K*_M_ constants for other phenolic compounds are considerably higher [24,34]. In our case, *Sl*Lac2 exhibited a preference (affinity 6-fold) for syringaldazine. By comparison, *Ps*Lac1 from the marine-derived fungus *Pestalotiopsis* sp., showed *K*_M_ values of 4 µM and 24 µM for syringaldazine and ABTS, respectively, while *Ps*Lac2, did not exhibit a specific preference for either substrate, with higher *K*_M_ values (hundreds of µM) [28]. Two other marine-derived fungal laccases from *Trematosphaeria*
*mangrovei* [29] and *Cerrena unicolor* [31] showed different behaviours, with *K*_M_ values of 1.42 mM and 54 μM against ABTS. Even though the affinity was stronger for ABTS, the catalytic efficiency (*K*_cat_/*K*_M_) of *Sl*Lac2 was slightly higher for syringaldazine (7.42 s^−1^ mM^−1^). This value is very close to the *K*_cat_/*K*_M_ measured for *Pestalotiopsis* sp. Lac2 (5.67 s^−1^ mM^−1^) [28]. *Ps*Lac2 was described as a relatively versatile laccase, exhibiting a similar catalytic efficiency for syringaldazine, ABTS and DMP (5.67, 4.33 and 9.1 s^−1^ mM^−1^, respectively). By contrast, the second laccase of *Pestalotiopsis* sp., *Ps*Lac1 showed the highest catalytic efficiency for syringaldazine at 82.93 s^−1^ mM^−1^, and a 3-times lower *K*_cat_/*K*_M_ for the other two substrates ABTS and DMP (29.25 and 24.53 s^−1^ mM^−^^1^, respectively). Higher catalytic efficiency for syringaldazine can be found in the literature, such as for the laccase-like enzyme cloned form a marine library with 29 s^−1^ mM^−1^ [23]. In conclusion for the kinetic parameters, *Sl*Lac2 was shown to possess the general characteristics of fungal laccases, with the highest efficiency for syringaldazine.

The physical and biochemical properties of the purified recombinant *Sl*Lac2 were tested for further comparisons with fungal laccases. As reported by Baldrian, fungal laccases typically exhibit an optimal pH in the acidic pH range (3.0–5.0), using ABTS as substrate [33], whereas the optimal pH for the oxidation of syringaldazine is higher (3.5–6.0) [33]. The purified recombinant *Sl*Lac2 was tested to determine its optimal pH in the range 3.0–9.0 using syringaldazine as a substrate, and it showed that the optimal pH value driving the maximum activity of the recombinant enzyme was pH 6.0, with a measurable activity at pH 8.0. For the laccases obtained from *Pestalotiopsis sp*. and *T. mangrovei*, the optimal pH was determined with ABTS as substrate: 5.0 and 4.0, respectively [28,29]. Usually, the stability of most fungal laccases is higher at acidic pH and decreases when close to neutral pH conditions, which is the case for the terrestrial *P. cinnabarinus* laccase [34]. In our work, we demonstrated that the purified recombinant *Sl*Lac2 was relatively resistance in several pH conditions tested (3.0 to 7.0). In addition, *Sl*Lac2 was more stable in the tested range of pH than *Pestalotiopis* sp. laccases, i.e., *Ps*Lac1 was less stable at alkaline pH. It showed 30% residual activity after 100 h of incubation at pH 6.0, whereas *Ps*Lac2, was instead more sensitive to low pH; its stability decreased with time between pH 3.0 and 6.0, and abruptly at pH 2.0 (no activity left after 40 min of incubation) [28].

Concerning the behaviour of fungal laccases with temperature, previous studies found that the optimal temperature was between 50 and 60 °C for most fungal laccases [32,33]. In our study, we also showed that the optimum temperature of the purified laccase *Sl*Lac2 was in this range (60 °C). Compared with marine laccases, *T. mangrovei* and *Ps*Lac 1 had slightly higher optimal temperatures: 65 °C (syringaldazine. 80 °C with ABTS) and 70 °C (ABTS), respectively [28,29]. Regarding thermal stability, *Sl*Lac2 was stable for 180 min at temperatures ranging from 25 to 50 °C, whereas the activity was significantly lost above 60 °C, as demonstrated for *Ps*Lac2 [28]. In addition, *T. mangrovei* laccase and *Ps*Lac1 showed curves with more pronounced thermal deactivation at temperatures above 60 °C [28,29,30]. We conclude that *Sl*Lac2 presents novel, useful properties due to its activity relative to temperature and pH, and to its stability compared with its marine laccase homologs.

Enzyme activity is often affected positively or negatively by metal ions present in the mixtures [35,36,37,38]. Because laccases are used in several processes containing metal ions, it is useful to characterise the effect of metal ions on their activities. It has been shown in many studies that laccase activities vary with the type of metal ion present and the enzyme source [38,39,40,41]. *Sl*Lac2 showed that adding a 10 mM concentration of Mo^2+^ and Ag^2+^ to the purified enzyme left 70% and 40% of the activity, respectively, but the presence of AsO4^3−^, Fe^2+^, Cd^2+^, Zn^2+^, and especially Cu^2+^, caused the loss of all or most activity. For instance, Cu^2+^ has already been shown to inhibit *Scytalidium thermophilum* laccase activity [42]. Among several metal ions tested, Xu et al. showed that Ag^+^, Ag^2+^, Li^+^, and Pb^2+^ could inhibit the laccase of *Cerenia* sp. reversibly. Hg^+^ was shown to reduce pH and thermal stability, and dynamic simulation showed the presence of binding sites for Hg close to copper binding sites on the laccase molecule [43]. For *T. mangrovei*, the effect of metal ions was analysed at 1 mM, and only Fe^2+^ showed a complete inactivation of its activity, whereas Cu^2+^ was not tested [29]. At the same concentration, Fe^3+^ showed a 30% inactivation of the marine-derived fungus, *C. unicolor* laccase, whereas Cu^2+^ addition produced a slight effect with a loss of less than 10% of activity [31]. The most common laccase inhibitors tested include cysteine, EDTA, sodium fluoride, sodium azide, dithiothreitol, thioglycolic acid, and diethyldithiocarbamic acid. These inhibitors are not laccase-specific and their applications on phenoloxidases was because they could inhibit metalloenzymes [44,45]. They can act on the active site: EDTA binds to copper atoms and stops the transfer of electrons. Cysteine, dithiothreitol, thioglycolic acid, and diethyldithiocarbamic acid are reducing substances described as potential laccase inhibitors sequestering dioxygen and stopping the oxidation of the phenolic substrates. However, Johannes and Majcherczyk (2002) have tested several potential laccase inhibitors including dithiothreitol, thioglycolic acid (TGA), diethyldithiocarbamic acid (DDC), cysteine, and sodium azide (NaN_3_) [46]. They found that only NaN_3_ stopped the substrate oxidation by no oxygen uptake, whereas the sulfydryl compounds did not affect the oxygen consumption and even increased it through an autoxidation reaction. *Sl*Lac2 was significantly inactivated by NaN_3_, l-cysteine and 2-mercaptoethanol, whereas *C. unicolor* laccase was inhibited by 1 mM of NaN_3_ (95% inhibition) and 2-mercaptoethanol (67% inhibition), but not by l-cysteine (0.1 mM) [28]. For *T. mangrovei* laccase, a 50% inhibition was demonstrated for 1 mM of NaN_3_ and the most efficient inhibitor was shown to be NaCN (80% imbibition) [29].

Biochemical reactions involving laccases in compatible organic solvents give access to some insoluble substrates, which may help in the detoxification of several sparingly water-soluble persistent organic pollutants. However, several studies have shown negative effects of organic solvents on laccase activity [47,48]. For instance, Robles et al. [49] studied a laccase of the hyphomycete *Chalara* (syn. *Thielaviopsis*) *paradoxa* CH32 and showed an activity decrease of 27% and 36% in the presence of 25% (vol/vol) ethanol and methanol, respectively. In our study, we showed that with 40% (vol/vol) of ethanol and methanol, there was a similar decrease in the activity from about 30% and 20%, respectively. However, of the solvents tested in our experiments, acetone was the most damaging, probably by precipitating the proteins. Both free and immobilised laccases decreased their activity with an increase in water-miscible organic solvent concentrations, and in most reported cases, immobilised laccase is less strongly affected than the free enzyme by the organic solvent concentrations, which could offer an alternative, depending on the solvents needed to implement for the laccase application.

As *Sl*Lac2 was isolated from a marine-derived fungus, this enzyme behaviour was studied in relation to saline conditions. In the presence of sea salt, *Sl*Lac2 was affected, with a continuous decrease in its activity following the sea salt concentration increase. For the two laccases of the marine-derived fungus *Pestalotiopsis* sp., *Ps*Lac1 and 2, completely different behaviour was observed, as their activities were enhanced in the presence of increasing concentrations of sea salts [28]. *Ps*Lac2 activity was clearly stronger (1.6 times higher compared with *Ps*Lac2 at 5% sea salt). In addition, *T. mangrovei* laccase lost half of its activity in only 1 mM NaCl [29] while *Pestalotiopsis* sp. lytic polysaccharide monoxidases A (LMPOA) could act on cellulose for up to 6% sea salt [50]. A few studies have been carried out on enzymes originating from saline environments to identify the molecular determinants responsible for their salt tolerance. Recently, Li et al. [51] demonstrated that two amino acid sites were involved in the salt activation, as Cl^−^ ion could bind to specific local sites to interfere with substrate binding and/or electron transfer. In addition, salt-adapted enzymes were shown to be strongly negatively charged on their surface, a characteristic contributing to protein stability and to enzyme activities adapted to extreme osmotic conditions [52,53,54]. *Sl*Lac2 was shown to possess a slightly higher positive charge distribution at the protein surface. In contrast, *Ma*Lac1 and *Ps*Lac1 exhibited a relatively even balance of negative charges over positive charges (D + E/R + K ratio), whereas *Ps*Lac2 presented a very high ratio of 3.92. In a previous study on LMPOs identified from *Pestalotiopsis* sp., the calculated ratio also exhibited very high ratios greater than 4.0 [50] and an excess of negatively charged residues at their surface, typical of marine enzymes. Altogether, these results show that the resistance to salt is not yet fully understood, and that further research is still needed to elucidate the adaptation mechanisms.

## 4. Materials and Methods

### 4.1. Strains and Culture Conditions

*Escherichia coli* strain TOP 10 was used for vector storage and propagation. *Aspergillus niger* strain D15#26 (pyrG deficient) [55] was used for the heterologous expression of the *Sl*Lac2 encoding synthetic gene. After co-transformation with vectors respectively containing the pyrG gene and the laccase cDNA, transformants of *A. niger* were grown for selection on solid minimal medium without uridine and containing 70 mM of NaNO_3_, 7 mM of KCl, 11 mM of KH_2_PO_4_, 2 mM of MgSO_4_, glucose 1% (wt/vol), and trace elements (1000 × stock; 76 mM ZnSO_4_, 178 mM H_3_BO_3_, 25 mM MnCl_2_, 18 mM FeSO_4_, 7.1 mM CoCl_2_, 6.4 mM CuSO_4_, 6.2 mM Na_2_MoO_4_, and 174 mM EDTA). For the screening procedure of the positive transformants, 100 mL of culture medium containing 70 mM NaNO_3_, 7 mM KCl, 200 mM Na_2_HPO_4_, 2 mM MgSO_4_, glucose 5% (wt/vol), and trace elements was inoculated with 2 × 10^6^ spores mL^−1^ in a 250 mL baffled flask.

### 4.2. Cloning and Expression of SlLac2-Encoding Gene

The open reading frame sequence encoding *Sl*Lac2 was synthesised and codon bias optimised for *A. niger* (GeneArt, Regensburg, Germany), with some modifications. The amino acids of the signal peptide, which was predicted with the program SignalP hosted on the ExPASy Proteomics server (http://www.expasy.ch), were replaced by the 24-amino-acid glucoamylase (GLA) preprosequence from *A. niger* (MGFRSLLALSGLVCNGLANVISKR). Two restriction sites (*BssH*II and *Hind*III) were respectively added at the 5′ and 3′ ends of the sequence for cloning into the expression vector pAN52.4 (GenBank/EMBL accession number Z32699). In the final expression cassette, the *Aspergillus nidulans* glyceraldehyde-3-phosphate dehydrogenase-encoding gene (*gpdA*) promoter, the 5′ untranslated region of the *gpdA* mRNA, and the *A. nidulans trpC* terminator were used to drive the expression of the inserted coding sequences. The co-transformation was carried out as described by Punt and van den Hondel [25] using both the expression vector containing the expression cassette and pAB4-1 [56] containing the *pyrG* selection marker in a 10:1 ratio. Transformants were selected for uridine prototrophy by growth on selective solid minimal medium (without uridine).

### 4.3. Screening of Transformants and Laccase Activity Assay

To screen for the best clones for enzyme production in liquid medium, 100 mL of minimal medium (adjusted to pH 5.5 with 1 M of citric acid) was inoculated with 2 × 10^6^ spores mL^−1^ in a 250 mL flask. The cultures were incubated for 10 days at 30°C in a shaker incubator (110 rpm), and pH was adjusted daily to 5.5 with 1 M citric acid. From these liquid cultures, aliquots (2 mL) were withdrawn daily, and mycelia were pelleted (20 min at 15,000× *g*). Laccase activity of the resulting supernatant was assayed spectrophotometrically by monitoring the oxidation of syringaldazine (1 mM) as substrate at 436 nm (ɛ_436_ = 29,300 M^−1^ cm^−1^) in a sodium acetate buffer (50 mM, pH 6.0). The reaction was monitored for 1 min at 30 °C in an Uvikon XS spectrophotometer (BioTek Instruments, Colmar, France). Activity is expressed in nkat mL^−1^, 1 nkat corresponding to the oxidation of 1 nanomole of substrate per second. Measurements in all the experiments were performed in triplicate.

### 4.4. Production and Purification of Recombinant SlLac2

For protein production, the best positive clone corresponding to the clone with the highest laccase activity was selected and cultured. 930 mL of culture medium containing 70 mM NaNO_3_, 7 mM KCl, 200 mM K_2_HPO_4_, 2 mM MgSO_4_, glucose 10% (wt/vol), trace elements and adjusted to pH 5.0 with a 1 M citric acid solution, inoculated with 2 × 10^6^ spores mL^−1^, was prepared to initiate a large-scale protein production (1 L). The culture was harvested after 11 days. The culture medium was clarified by filtration through GF/D, GF/A and GF/F glass fibre filters (Whatman, Maidstone, UK), followed by filtrations through 0.45 μm and 0.22 μm polyethersulphone membranes (Express Plus, Merck Millipore). The collected filtrate was concentrated by ultrafiltration through a polyethersulphone membrane with a 10 kDa molecular mass cut-off (Vivaflow crossflow cassette, Sartorius, Les Ulis, France). After dialysing overnight at 4 °C against 100 mM Tris buffer pH 8.0, 150 mM NaCl and 1 mM EDTA, the sample was loaded onto on a CM-Sepharose fast flow column (GE Healthcare Life Science, Velizy-Villacoublay, France) using an AKTA purifier (GE Healthcare Life Science). The sample was loaded onto the column previously equilibrated with a binding buffer (0.2 mM sodium tartrate buffer pH 5.0, containing 25 mM NaCl). Protein elution was performed using a linear gradient of 0–100% of the elution buffer (0.2 mM of sodium tartrate buffer pH 5.0, containing 1 mM NaCl). Activity was determined in the 10 mL collected fractions, and protein production was evaluated by SDS-PAGE. The active fractions were concentrated through a 10 kDa molecular mass cut-off Amicon membrane (Millipore). The concentrated samples were loaded onto a Sephacryl S-200HR size exclusion chromatography column (GE Healthcare Life Science) equilibrated with a 50 mM sodium acetate buffer pH 5.0, containing 50 mM NaCl. 5 mL fractions were collected and assayed for laccase activity as described above, and protein homogeneity was tested by SDS-PAGE. For the different purification steps, the total protein concentration was determined with the Bradford assay using the BioRad Protein Assay Kit (BioRad, Marnes-la-Coquette, France) and bovine serum albumin (BSA) as a standard. Final protein concentration was determined spectrophotometrically at 280 nm using a NanoDrop 2000 (Thermo Fisher Scientific, Illkirch, France) and the theoretical molar extinction coefficients *Sl*Lac2 at 280 nm, 125,290 M^−1^ cm^−1^. The molecular mass of the purified protein was determined by 12% sodium dodecyl sulphate-polyacrylamide gel electrophoresis (SDS-PAGE).

### 4.5. Bioinformatic Analysis

The molecular mass, theoretical pI, and molar extinction coefficient of enzymes were predicted by the ProtParam tool (http://web.expasy.org/protparam/). Protein sequences were aligned using MUSCLE [57,58] and CLUSTAL W [59] at http://www.ebi.ac.uk/Tools/msa/. Signal peptides were predicted using Signal P [60] at http://www.cbs.dtu.dk/services/SignalP/. N-glycosylation sites [60] were predicted at http://www.cbs.dtu.dk/services/NetNGlyc/. The automated protein structure homology modelling online tool Phyre 2 [61] was used to predict the three-dimensional models of *Sl*Lac2 using the closest homologs of known structure available in the Protein Data Bank (PDB). Protein models c2q9oA (*Melanocarpus albomyces* laccase, 44% identity), c5lwxA (*Aspergillus niger* laccase McoG H253D variant, 37% identity), c3ppsD (*Thielavia arenaria* laccase, 45% identity) and c3sqrA (*Botrytis aclada* laccase, 44% identity) were used as templates for *Sl*Lac2. Surface charges were calculated, and all the figures were prepared with PyMOL.

### 4.6. Determination of N-Terminal Amino Acid Sequence and Glycosylation Level

The N-terminal sequence was determined according to Edman degradation. Analysis was carried out on an Applied Biosystem 476Asequencer by the proteomic platform of the Institut de Microbiologie de la Méditerrranée, CNRS, Aix-Marseille Université (France). Matrix-assisted laser desorption ionisation—time-of-flight mass spectrometry of samples was carried out on a Microflex II time-of-flight mass spectrometer (Bruker Daltonik, Germany).

To determine the level of glycosylation of the purified laccase *Sl*Lac2, a deglycosylation reaction was carried out using the PNGase F (New England Biolabs, France). 15 µg of purified laccase was mixed with 4 µL of the denaturation solution (5% SDS and 40 mM dithiothreitol (DTT)) denatured at 100 °C for 10 min. The denatured sample was deglycosylated in a final volume of 40 µL containing 1% Nonidet P-40, 50 mM sodium phosphate buffer (pH 7.5) and 2 µL of the deglycosylation enzyme PNgase F solution (500,000 units mL^−1^). The reactions were incubated at 37 °C for 2 h. After the deglycosylation, 10 µL of the digested enzyme was loaded onto a 12% homogeneous SDS-PAGE gel.

### 4.7. Substrate Specificity and Kinetics

Enzyme activities were measured using a UVIKONxs spectrophotometer (Bio-TEK Instruments) at 30 °C by following the oxidation of different aromatic substrates: 2,2′-azino-bis(3-ethylbenzothiazoline-6-sulphonic acid) (ABTS), 2,6-dimethoxyphenol (DMP) and syringaldazine. The absorbance increases at 436 nm (ɛ_436_ = 29,300 M^−1^ cm^−1^), 469 nm (ɛ_469_ = 27,500 M^−1^ cm^−1^) and 530 nm (ɛ_530_ = 65,000 M^−1^ cm^−1^) were followed for ABTS, DMP, and syringaldazine, respectively. Kinetic constants (enzyme concentration 0.18 mg mL^−1^) were determined for each of these compounds (substrate range of 0.05 to 1.1 mM for ABTS, and 0.025 to 0.1 mM for DMP and syringaldazine) with sodium acetate buffer (50 mM, pH 4.0, 5.0, and 6.0, respectively). All assays were performed in triplicate. Mean apparent affinity constant (Michaelis constant, *K_m_*) and enzyme turnover (catalytic constant, *k_cat_*) values and standard errors were obtained by nonlinear least-squares fitting to the Michaelis-Menten model. Fitting of these constants to the normalized Michaelis-Menten equation υ = (*k_cat_*/*K_m_*)[S]/(1 + [S]/*K_m_*) yielded enzyme efficiency values (*k_cat_*/*K_m_*) with their standard errors

### 4.8. Effect of pH and Temperature on the Activity and Stability of the Laccase

The effect of pH on the purified enzyme was determined by measuring the enzyme activity at 30 °C in various buffers at different pH (2.5–10.0) using the following buffers: 50 mM sodium acetate buffer (pH 2.5–7.0), phosphate buffer (pH 7.0–9.0), and sodium bicarbonate (pH 9.0–10.0), against 1 mM syringaldazine as substrate at 30°C. The pH stability of the purified enzyme was determined by incubating the enzyme in the different buffer solutions (50 mM) with different pH values (3.0–7.0) and allowing the mixture to stand for 4 h, 24 h and 48 h incubation at room temperature. The residual enzyme activity was determined under standard assay conditions after centrifugation to remove the denatured proteins.

To determine the effect of temperature on laccase, the activity of the purified enzyme was measured in a sodium acetate buffer (50 mM, pH 6.0) at different temperatures ranging from 20 to 80°C. The stability to thermal treatment of purified enzyme was determined by measuring residual activity after incubation at different temperatures (25–70°C) and pH 6.0 for 30 min to 3 h. All the reactions were run in triplicate.

### 4.9. Effect of Metal Ions, Inhibitors, Solvents, and Sea Salt on Laccase Activity.

To study the effect of metal ions, potential inhibitors, different solvents, and sea salt on the purified enzyme, the activity was determined in the presence of metal ions in standard laccase activity conditions. The following compounds were used as the source of metal ions: CuSO_4_.5H_2_O, Na_2_MoO_4_, FeSO_4_.7H_2_O, CdCl_2_, Na_3_AsO_4_.12H_2_O, and ZnSO_4_.7H_2_O. They were added at a final concentration of 10 mM. The potential inhibitors were l-cysteine, 2-mercaptoethanol, EDTA, SDS, and sodium azide (final concentrations of 0.5, 0.05 and 0.05 mM). The different organic solvents were ethanol, methanol, iso-propanol, glycerol, and acetone at different concentrations: 10, 20, and 40%. Different concentrations of sea salt (Sigma-Aldricht, Lyon, France) (1–5% wt/vol) were added to the reaction mixture and the activity was determined in standard conditions. All the reactions were run in triplicate.

### 4.10. Nucleotide Sequence Accession Number

The gene sequences encoding *Sl*Lac2 were deposited in the nucleotide sequence database (GenBank) under the accession code MT470191.

## 5. Conclusions

The present work carried out on the *S. lucomagnoense* gives new insights into the marine-derived laccases, which are catalysts of great interest for many biotechnological applications. In addition, *Sl*Lac2 produced in *A. niger* presented potential properties related to pH and some tolerance to sea salt, which could be exploited in biotechnological applications, such as the pulp and paper sector and enzymatic degradation of industrial dyes, for which alkaline-active enzymes and high tolerance to salt are needed, respectively

## Figures and Tables

**Figure 1 ijms-21-08402-f001:**
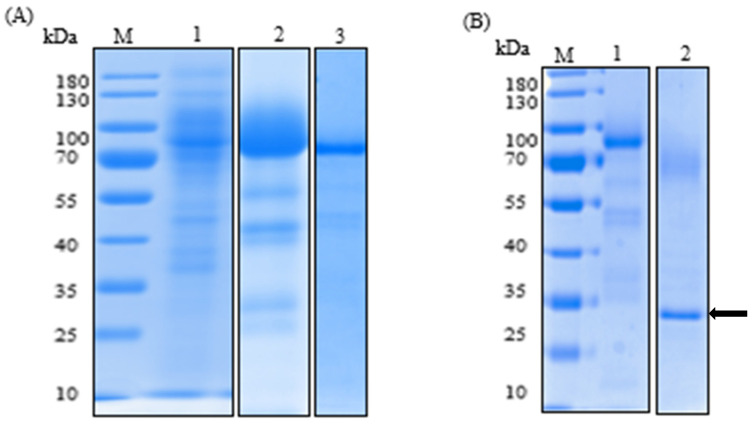
Sodium dodecyl sulfate-polyacrylamide gel electrophoresis (SDS-PAGE) (12% polyacrylamide gels) and N-glycosylation analysis of the purified *Sl*Lac2. Three-step purification procedure of *Sl*Lac2 (**A**). Lanes 1, culture supernatant concentrated by ultrafiltration; 2, purified *Sl*Lac2after carboxymethyl (CM)-Sepharose column; 3, purified *Sl*Lac2 obtained after Sephacryl S-200HR. N-glycosylation analysis by SDS-PAGE of the purified *Sl*Lac2 (**B**). Lanes 1, purified *Sl*Lac2; 2; purified *Sl*Lac2 deglycosylated by N-glycosidase PNGase F (marked with an arrow). M are molecular mass standards. Proteins were stained with Coomassie blue.

**Figure 2 ijms-21-08402-f002:**
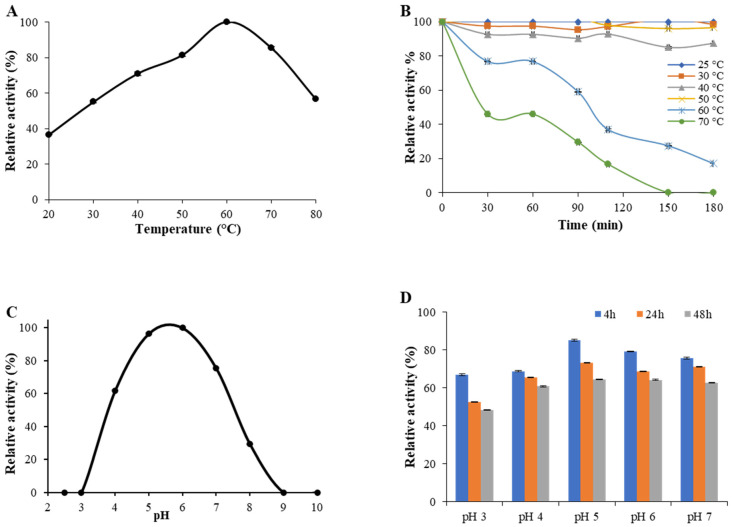
Enzyme activity and stability of *Sl*Lac2 at different pH and temperature. (**A**) The oxidation of syringaldazine (1 mM) was determined for the temperature curve in a sodium acetate buffer (50 mM, pH 6.0), and (**C**) for the pH curve at 30 °C. Values were calculated as a percentage of maximum activity (set at 100%) at optimum temperature and pH. (**B**) Residual activities were estimated after 30, 60, 90, 120, and 180 min of incubation at six different temperatures ranging from 25 to 70 °C or (**D**) after 4, 24, and 48 h at five different pH values ranging from 3.0 to 7.0. Residual activities are expressed as a percentage of the initial activity (point at time 0, measured immediately after adding the enzyme), which was set at 100%. Assays were performed using syringaldazine as a substrate in standard conditions. Each data point (mean +/− standard deviation) is the result of triplicate experiments.

**Figure 3 ijms-21-08402-f003:**
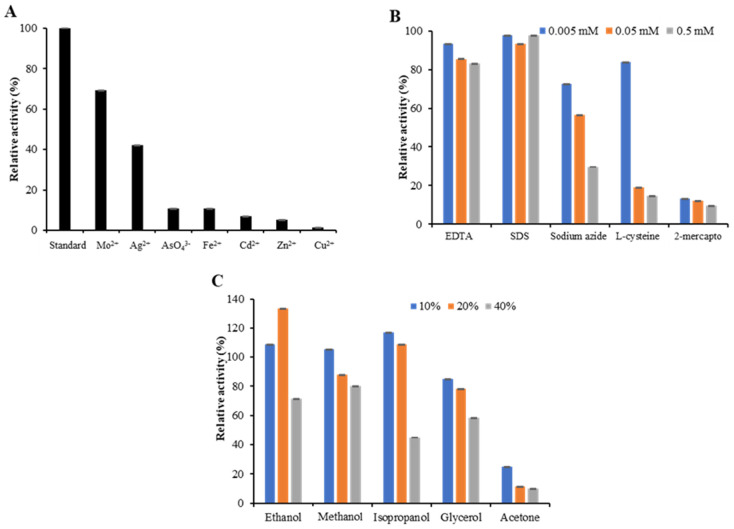
Effect of metal ions, inhibitors, and solvents on *Sl*Lac2 activity. (**A**) Effects of various metal ions (Mo^2+^, Ag^2+^, AsO_4_^3−^, Fe^2+^, Cd^2+^, Zn^2+^, or Cu^2+^) at 10 mM on the purified recombinant enzyme. (**B**) Effect of various inhibitors (EDTA, SDS, 2-mercaptoethanol, l-cysteine, and sodium azide) on *Sl*Lac2 activity at different concentrations (0.5, 0.05 and 0.005 mM). (**C**) Effect of solvents (ethanol, methanol, isopropanol, glycerol, and acetone) at different concentrations (10, 20, and 40% vol/vol) on *Sl*Lac2 activity.

**Figure 4 ijms-21-08402-f004:**
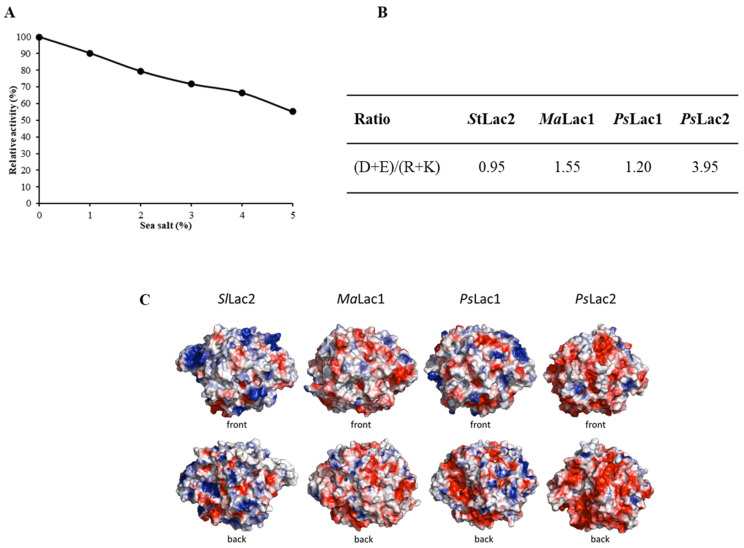
Effect of sea salt and surface charge of *Sl*Lac2. (**A**) Effect of different concentrations (1, 2, 3, 4 and 5% wt/vol) of sea salt on *Sl*Lac2 activity. (**B**) (D + E)/(R + K) amino acids ratio of *Sl*Lac2 compared with those from *Melanocarpus albomyces* laccase 1, *Ma*Lac1 (Q70KY3) and *Pestalotiopsis* sp. laccases 1 and 2, *Ps*Lac1 (KY554800) and *Ps*Lac2 (KY5548001). (**C**) Surface charge plots (negative and positive charges are in red and blue, respectively) of *Sl*Lac2 compared with those from *M. albomyces* laccase 1, *Ma*Lac1 and *Pestalotiopsis* sp. laccases 1 and 2, *Ps*Lac1 and *Ps*Lac2. The surface potentials were calculated using the vacuum electrostatics function of the PyMOL molecular graphics system (Schrödinger, New York, NY, USA).

**Table 1 ijms-21-08402-t001:** Properties of the putative laccases of *Stemphylium lucomagnoense* and the closest characterised laccases from the terrestrial fungus, *Melanocarpus albomyces* (*Ma*Lac1: Q70KY3), and the marine-derived fungus *Pestalotiopsis* sp. KF079 (*Ps*Lac1: KY554800 and *Ps*Lac 2: KY554801), including the number of amino acids of the signal peptide and the mature protein, the calculated molecular mass and pI, and the amino acids involved in the enzyme coordination sites for the Type 1, 2 and 3 coppers. APMZ2_prot14771 * encoding *Sl*Lac2 was deposited in the nucleotide sequence database (GenBank) under the accession code MT470191.

Accession Number	Signal Peptide (Amino Acids)	Mature Protein (Amino Acids)	Molecular Mass (Da)	PI	Coordination Sites
Q70KY3	22	601	68958	5.18	IHWHG WYHSH HPMHLH HCHIAWH
KY554800	21	555	60904	6.09	IHWHG WYHSH HPMHLH HCHIAWH
KY554801	18	543	59813	4.10	IHWHG WYHSH HPMHLH HCHIAWH
APMZ2_prot14771 *	21	592	64581	8.07	IHWHG WYHSH HPIHLH HCHIAFH
APMZ2_prot8345	23	578	63456	5.64	IHWHG WYHSH HPIHLH HCHIAWH
APMZ2_prot7177	21	615	68221	6.03	IHFHG WYHSH HPIHKH HCHINNH
APMZ2_prot10330	No	754	83453	5.40	IHFHG WYHAH HPFHLH HCHNMWH
APMZ2_prot15523	No	729	80156	5.37	LHAHG WYHSH HPMHLH HCHLAWH
APMZ2_prot2393	No	554	60438	4.66	LHFHG WYHSH HPFHLH HCHIEWH
APMZ2_prot9198	19	576	64532	5.08	MHWHG FYHSH HPFHLH HCHVLQH

**Table 2 ijms-21-08402-t002:** Purification for the recombinant *Sl*Lac2 produced in *Aspergillus niger* D15#26. CM-Sepharose: carboxymethyl-Sepharose.

Purification Steps	Volume (mL)	Total Activity (nKat)	Protein (mg)	Specific Activity (nkat mg^−1^)	Activity Yield (%)	Purification Factor (fold)
Culture medium	930	78,027	176	443	100	1
Ultrafiltration	500	99,000	156	635	126.9	1.4
CM-Sepharose	325	20,775	15.4	1349	26.6	3.0
Sephacryl S-200HR	40	18,782	7.2	2609	24.1	5.9

**Table 3 ijms-21-08402-t003:** Kinetic parameters of *Sl*Lac2.

Substrate		*K*_M_ (mM)	*K*_cat_ (s^−1^)	*K*_cat_/*K*_M_ (s^−1^ mM^−^^1^)
ABTS	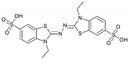	0.0206 +/− 0.0039	0.153 +/− 0.0026	7.42
DMP	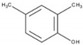	0.0240 +/− 0.0039	0.0059 +/− 0.0003	0.24
Syringaldazine	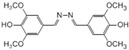	0.0035 +/− 0.00057	0.0074 +/− 0.0002	2.11

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
