# Peer review of "Enzyme Properties of a Laccase Obtained from the Transcriptome of the Marine-Derived Fungus Stemphylium lucomagnoense"

_ijms, 2020, doi:10.3390/ijms21218402_

Round 1

Reviewer 1 Report

The manuscript looks fine except for the following.

Table :.It is not necessary show the total activity and specific activity with decimal points because the values are very large,

Please show standard deviation for data in Fig 2 A and C,Fig 3A and Fig 4A.

Do the data in Fig 3B and Fig 3C show standard deviations or are the standard deviations too small to be seen.

Author Response

We would like to thank Reviewer 1 for its suggestions and recommendations which we believe have resulted in an improved manuscript. You will find a point by point response to all remarks raised by the reviewers.

Table 2: It is not necessary show the total activity and specific activity with decimal points because the values are very large

  • Corrected

Please show standard deviation for data in Fig 2 A and C, Fig 3A and Fig 4A.

Do the data in Fig 3B and Fig 3C show standard deviations or are the standard deviations too small to be seen.

  • All the standard deviations are marked on the figures but they are too small to be seen.

Reviewer 2 Report

Detailed remarks to the manuscript

  1. Page 3, Table 1, column 4 (Molecular mass).

Authors are asked to check if they put the values of molecular mass in proper units. In my opinion they are not in (kDa) but in (Da)

  1. Page 4, line 13 – ‘Purification of the recombinant laccase and study of physical-chemical properties’

I suggest to change this phrase to: ‘Purification of the recombinant laccase and study of its physico-chemical properties’

  1. Page 5, Line 28, Figure 1 – the resolution is too low.

I Would like to ask Authors to improve the resolution of the Figure 1.

  1. Page 5, line 31 ‘purification step’

I suggest to change to ‘purified S/Lac2 obtained’

  1. Page 5, lines 46 – 51, Subsection 2.3. Kinetic parameters

This subsection should be obligatory improved. It should be complemented by Michaelis-Menten or other graphs that present the results of performed kinetic studies. There is no information about the concentration of enzyme used during the studies as well as the range of the concentration of each substrate.

  1. Page 5, lines 49 – 50 (KM = 3.5 mM) … (KM = 20.6 mM) …. (KM = 24.4 mM) and Page 6, Table 2 KM (mM)

Authors are asked to unify the KM unit to μM or mM, it should be the same both in the text and the Table 2.

  1. Page 6, line 53, Table 2

Authors are asked to put mean +/- standard deviation to obtained values of kinetic parameters

  1. Page 6, line 64 ‘residual’

Authors are asked to change it to ‘initial’

  1. Page 7, Figure 2

The error bars of mean +/- standard deviation are missing on Fig.2A, B and C.

  1. Page 7, line 94 – Page 8 line 95 ‘Of all the organic solvents tested, acetone showed a markedly negative effect on SlLac2 activity, causing more than 80% activity inhibition at a concentration…’

Authors are asked to explain if this was the activity inhibition for sure or they have in mind the activity inactivation? Please write if this phenomenon was reversible or irreversible.

  1. Page 8, lines 101 – 103 ‘Effect of different concentrations of solvent (ethanol, methanol, isopropanol, glycerol, and acetone) on SlLac2 activity at different concentrations (10, 20 and 40% vol/vol).’

Authors are asked to change this sentence to: ‘Effect of solvents (ethanol, methanol, isopropanol, glycerol, and acetone) at different concentrations (10, 20 and 40% vol/vol) on SlLac2 activity.’

  1. Page 9, Figure 3A

Authors are asked to put the error bars to the graph.

  1. Page 9, Figure 3B

Authors are asked to explain the abbreviations: D, E, R, K.

  1. Page 9, line 119, Caption of Fig.3B

There is mistake in the description of Fig.3B. In my opinion the current one is the description of Fig.3C and the right description of Fig.3B is missing. Authors are asked to correct these descriptions.

  1. Page 9, line 125 ‘programme’

I suggest to change this word to: ‘project’

  1. Page 10, lines 149 ‘published’

I suggest to change this word to: ‘literature reports on’

  1. Page 10, line 159

Please insert the word ‘affinity’ before ‘6-fold

  1. Page 11, line 207 ‘disappearance’

I suggest to change this word to: ‘loss’

  1. Page 11, line 237 ‘of’

I suggest to change this word to: ‘from’

  1. Page 11, line 243 ‘its’

I suggest to change this word to: ‘this’

  1. Page 12, line 248 ‘1.6 times …’

Please insert the word ‘higher” after ‘1.6 times …’

  1. Page 12, lines 254,255

Please check the text editing.

  1. Page 14, lines 357 – 364, Subsection ‘Substrate specificity and kinetics’

This subsection should be obligatory revised. Among others, Authors are asked to insert the concentration range of the individual substrates used in kinetic studies as well as the concentration of applied enzyme (S/Lac2). In current manuscript the information about the method/tools for determination of kinetic parameters is also missing. I would like to ask if the kinetic experiments were performed in triplicate?  

  1. Page 14, line 377 ‘…at different temperatures (0–70°C) and pH 6 for 30 min to 3 h’

Please change to: ‘..at different temperatures (25–70°C) and pH 6 from 30 min to 3 h

  1. Page 14, line 381 ‘The following substances were used for metal ions’

Please change to: ‘The following compounds were used as the source of metal ions’

  1. Page 15, line 394

Authors are asked to reject the phrase: ‘enzymes, and more specifically’

  1. Page 15, lines 392 – 397, Subsection Conclusions

The Conclusions section is too general.. It should be improved by providing the information about the significance of obtained results in a broader perspective. For example, what is their impact on the industrial sector, waste management or the environmental protection, etc.

Author Response

We would like to thank you reviewer 2 for its suggestions and recommendations which we believe have resulted in an improved manuscript. You will find a point by point response to all remarks raised by the reviewers.

Reviewer 2 :

  1. Page 3, Table 1, column 4 (Molecular mass).

Authors are asked to check if they put the values of molecular mass in proper units. In my opinion they are not in (kDa) but in (Da)

  • The reviewer 2 was right, they are in da, this was corrected in the Table 1.

  1. Page 4, line 13 – ‘Purification of the recombinant laccase and study of physical-chemical properties’

I suggest to change this phrase to: ‘Purification of the recombinant laccase and study of its physico-chemical properties’

  • Corrected as suggested. The term was also corrected page 17, line 60.

  1. Page 5, Line 28, Figure 1 – the resolution is too low.

I Would like to ask Authors to improve the resolution of the Figure 1.

  • We are sorry but this is not a problem of resolution but the fact that the fungal proteins are highly glycosylated and that some of them are appearing with smears and the image seems not clear. But you can see that the standards are really clear.

  1. Page 5, line 31 ‘purification step’

I suggest to change to ‘purified S/Lac2 obtained’

  • Corrected as suggested

  1. Page 5, lines 46 – 51, Subsection 2.3. Kinetic parameters

This subsection should be obligatory improved. It should be complemented by Michaelis-Menten or other graphs that present the results of performed kinetic studies. There is no information about the concentration of enzyme used during the studies as well as the range of the concentration of each substrate.

  • The concentration range of each substrate and the concentration of enzyme were added in the section 4.7. As suggested, the graph was loaded as additional data (S2) on the web site.

  1. Page 5, lines 49 – 50 (KM = 3.5 mM) … (KM = 20.6 mM) …. (KM = 24.4 mM) and Page 6, Table 2 KM (mM)

Authors are asked to unify the KM unit to μM or mM, it should be the same both in the text and the Table 2.

  • This was corrected in the text.

  1. Page 6, line 53, Table 2

Authors are asked to put mean +/- standard deviation to obtained values of kinetic parameters

  • The standard deviations were added in Table 2.0

  1. Page 6, line 64 ‘residual’

Authors are asked to change it to ‘initial’

  • Corrected

  1. Page 7, Figure 2

The error bars of mean +/- standard deviation are missing on Fig.2A, B and C.

  • The error bars were also too low to be seen, but they are marked in the figures.

  1. Page 7, line 94 – Page 8 line 95 ‘Of all the organic solvents tested, acetone showed a markedly negative effect on SlLac2 activity, causing more than 80% activity inhibition at a concentration…’

Authors are asked to explain if this was the activity inhibition for sure or they have in mind the activity inactivation? Please write if this phenomenon was reversible or irreversible.

  • This was not tested in our experiment and we could not answer the question.

  1. Page 8, lines 101 – 103 ‘Effect of different concentrations of solvent (ethanol, methanol, isopropanol, glycerol, and acetone) on SlLac2 activity at different concentrations (10, 20 and 40% vol/vol).’

Authors are asked to change this sentence to: ‘Effect of solvents (ethanol, methanol, isopropanol, glycerol, and acetone) at different concentrations (10, 20 and 40% vol/vol) on SlLac2 activity.’

  • Done

  1. Page 9, Figure 3A

Authors are asked to put the error bars to the graph.

  • The error bars were also too low to be seen, but they are marked in the figures.

  1. Page 9, Figure 3B

Authors are asked to explain the abbreviations: D, E, R, K.

  • The letters are international letters for amino acids. The term amino acid was added in the legend to clarify this.

  1. Page 9, line 119, Caption of Fig.3B

There is mistake in the description of Fig.3B. In my opinion the current one is the description of Fig.3C and the right description of Fig.3B is missing. Authors are asked to correct these descriptions.

  • The Reviewer is right, the figure legend was corrected as legend B was missing.

  1. Page 9, line 125 ‘programme’

I suggest to change this word to: ‘project’

  • Done

  1. Page 10, lines 149 ‘published’

I suggest to change this word to: ‘literature reports on’

  • Done

  1. Page 10, line 159

Please insert the word ‘affinity’ before ‘6-fold’

  • Done

  1. Page 11, line 207 ‘disappearance’

I suggest to change this word to: ‘loss’

  • Done

  1. Page 11, line 237 ‘of’

I suggest to change this word to: ‘from’

  • Done

  1. Page 11, line 243 ‘its’

I suggest to change this word to: ‘this’

  • Corrected

  1. Page 12, line 248 ‘1.6 times …’

Please insert the word ‘higher” after ‘1.6 times …’

  • Done

  1. Page 12, lines 254,255

Please check the text editing.

  • Yes, there was an error of editing, this was corrected.

  1. Page 14, lines 357 – 364, Subsection ‘Substrate specificity and kinetics’

This subsection should be obligatory revised. Among others, Authors are asked to insert the concentration range of the individual substrates used in kinetic studies as well as the concentration of applied enzyme (S/Lac2). In current manuscript the information about the method/tools for determination of kinetic parameters is also missing. I would like to ask if the kinetic experiments were performed in triplicate?

  • We added the missing information in this section and the standard deviations were added in Table 2.

  1. Page 14, line 377 ‘…at different temperatures (0–70°C) and pH 6 for 30 min to 3 h’

Please change to: ‘..at different temperatures (25–70°C) and pH 6 from 30 min to 3 h

  • Corrected

  1. Page 14, line 381 ‘The following substances were used for metal ions’

Please change to: ‘The following compounds were used as the source of metal ions’

  • Corrected
  1. Page 15, line 394

Authors are asked to reject the phrase: ‘enzymes, and more specifically’

  • Corrected
  1. Page 15, lines 392 – 397, Subsection Conclusions

The Conclusions section is too general.. It should be improved by providing the information about the significance of obtained results in a broader perspective. For example, what is their impact on the industrial sector, waste management or the environmental protection, etc.

  • We tried to improve the conclusion as suggested.

Reviewer 3 Report

The above mention current manuscript is well written and the presented topic (enzyme properties of a laccase) is interesting for publication in the journal: International Journal of Molecular Science.

 However, I have some comments and questions:

Materials and methods

Ɛ436 = 29,300 M-1 cm-1) in a sodium acetate buffer (50 mM, pH 6)…in whole document it should be 6.0

….. trace elements and adjusted to pH 5 with a… in whole document it should be 5.0

different pH (2.5–10)… 2.5–10.0…etc.

The explanation of abbreviation should be explained first time, when the authors use it, for examle abbreviation SDS-PAGE or Matrix-assisted laser desorption ionisation – time-of-flight MS

4.8. Effect of pH and temperature on the activity and stability of the laccase – there is not written how many times the samples were measured. Duplicate, triplicate etd…

Results, discussion

In the Figure 2 A, B, C the standard deviations are missing.

Laccase-producing transformants were identified by the appearance of a green zone around the colonies after 3 to 6 days of culture… There is no photo of this result, why?

The mention of figure 4 in the text of manuscript is missing.

A few studies have been carried out on enzymes originating from saline environments to identify the molecular determinants responsible for their salt tolerance.“ Which one? The references are missing.

Discussion in quite poor. It is more summary than discussion. Better explanation of results is needed.

Author Response

We would like to thank you and the reviewer 3 for his suggestions and recommendations which we believe have resulted in an improved manuscript. You will find a point by point response to all remarks raised by the reviewers.

Reviewer 3 :

Materials and methods

Ɛ436 = 29,300 M-1 cm-1) in a sodium acetate buffer (50 mM, pH 6)…in whole document it should be 6.0

….. trace elements and adjusted to pH 5 with a… in whole document it should be 5.0

… different pH (2.5–10)… 2.5–10.0…etc.

  • Corrected

The explanation of abbreviation should be explained first time, when the authors use it, for example abbreviation SDS-PAGE or Matrix-assisted laser desorption ionisation – time-of-flight MS

  • Corrected

4.8. Effect of pH and temperature on the activity and stability of the laccase – there is not written how many times the samples were measured. Duplicate, triplicate etd…

  • All assays were performed in triplicate, this was added at the end of Section 4.8

Results, discussion

In the Figure 2 A, B, C the standard deviations are missing.

  • As indicated previously, there are not missing. It is just that are very low.

…Laccase-producing transformants were identified by the appearance of a green zone around the colonies after 3 to 6 days of culture… There is no photo of this result, why?

  • Due to the number of figures, we did not show the related photos which correspond to minor information.

The mention of figure 4 in the text of manuscript is missing.

  • This was corrected in the all section 2.8.

A few studies have been carried out on enzymes originating from saline environments to identify the molecular determinants responsible for their salt tolerance.“ Which one? The references are missing.

  • The studies are described in the following lines, such as references 51, 52, 53 and 54.

Discussion in quite poor. It is more summary than discussion. Better explanation of results is needed.

  • We believe that our results have been appropriately compared to the existing literature on the subject and we have tried to summarize the key points for each paragraph.